# Content Analysis of Official Public Health Communications in Ontario, Canada during the COVID-19 Pandemic

**DOI:** 10.3390/ijerph21030351

**Published:** 2024-03-15

**Authors:** Maya Fields, Kelsey L. Spence

**Affiliations:** Department of Population Medicine, Ontario Veterinary College, University of Guelph, Guelph, ON N1G 2W1, Canada; fieldsm@uoguelph.ca

**Keywords:** risk communication, public health, COVID-19, content analysis

## Abstract

Effective communication by governmental organizations is essential to keep the public informed during a public health emergency. Examining the content of these communications can provide insight into their alignment with best practices for risk communication. We used content analysis to determine whether news releases by the Ontario government contained key elements of effective risk communication, as outlined by the Health Canada and Public Health Agency of Canada Strategic Risk Communication Framework. News releases between 25 January 2020 and 31 December 2022 were coded following the five elements of the framework: situational transparency, stakeholder-centered content; evidence-based rationales for decisions; continuous improvements in updating information; and descriptions of risk management. All 322 news releases contained at least one element of the framework, and all five elements were identified at least once across the dataset. Risk management, transparency, and stakeholder-centered content were the most frequently identified elements. News releases near the beginning of the pandemic contained most elements of the framework; however, over time, there was an increase in the use of vague language and lack of evidence-based rationales. Increasing transparency regarding evidence-based decisions, as well as changes in decisions, is recommended to improve risk communication and increase compliance with public health measures.

## 1. Introduction

Since early 2020, countries and governments worldwide have engaged in numerous communication strategies to update the public on the rapidly changing situation relating to the Coronavirus Disease 19 (COVID-19) pandemic. These strategies must be effective to educate the public on their risk of disease and increase compliance with any public health measures that are put in place. Risk communication, defined by the World Health Organization as “the real-time exchange of information, advice, and opinions between experts or officials and people who face a threat” [1], is essential to allow the public to make informed decisions to mitigate personal risks and implement preventative measures. In Canada, recommendations for preventative public health measures can vary across the country, as each province or territory has partial jurisdiction over healthcare decisions. As such, it is important that the public is provided with accurate, timely, and transparent communications to allow for effective decision-making within each jurisdiction.

Several frameworks have been previously created to guide effective risk communication by public health officials, including the Crisis and Emergency Risk Communication Emergency Risk Communication (CERC) framework [2], created by the Centers for Disease Control and Prevention (CDC), and the Health Canada and Public Health Agency of Canada Strategic Risk Communication Framework and Handbook [3]. The Strategic Risk Communication Framework was developed to help federal government scientists and communicators conduct risk communication in a more systematic and effective manner [3]. The framework emphasizes that communication is one of the most powerful influences on how people make decisions regarding health behaviors, and, therefore, providing comprehensive information to the public will allow them to make informed decisions regarding their health and wellbeing [3].

Despite the existence of established risk communication frameworks, it is unclear whether these methods are routinely applied in practice. Several studies have highlighted gaps in the application of best practices for risk communication during the COVID-19 pandemic. For example, a study examining how closely COVID-19 communications delivered by Scott Morrison, the Prime Minister of Australia, followed the CERC framework found that some components of the CERC model were not included in these communications [4]. Similarly, another study examining COVID-19 crisis communication on social media using constructs from the health belief model and extended parallel processing model found an overall low use of these constructs in captions and images [5]. The effective implementation of risk communication frameworks is further complicated by factors such as trust, emotions, and divergence in perspectives, which can affect how the public views the message [6,7,8]. Since the public judges the trustworthiness of communications based on the characteristics of their content, including consistency, repetition, timeliness, transparency, and uncertainty, these factors are often included as key elements within established risk communication frameworks [2,3,7]. However, further research on whether these frameworks are used in practice, especially within a Canadian context, is warranted.

The objective of this study was to determine whether the communications by the Ontario government contained key elements of effective risk communication, as defined by the guiding principles of the Health Canada and Public Health Agency of Canada Strategic Risk Communication Framework and Handbook [3]. These guiding principles outlined five key elements that were essential for risk communication between the government and the public. By identifying the presence or absence of these key elements among the communications, we aimed to determine whether there could be improvements in how COVID-19 risk could be communicated to the public.

## 2. Materials and Methods

Publicly available news releases from the Ontario government (Ministry of Health and the Office of the Premier) were obtained from the Ontario Newsroom (Government of Ontario, 2020–2022, https://news.ontario.ca/en (accessed on 8 November 2022)). Although the Ontario government does utilize other public engagement tools (e.g., Government of Ontario Announcements YouTube Channel), we obtained news releases from this source as it was where information was first released and upon which other statements were based, as well as the most extensive and frequently updated source available. All releases that included significant mention of COVID-19 (i.e., the main purpose of the communication was regarding COVID-19) between 25 January 2020 (the first news release regarding COVID-19) and 15 December 2022 (the final news release regarding COVID-19 from 2022) were included. News releases were copied verbatim from the website and imported into NVivo (NVivo 1.7.1, Lumivero, London, UK) for analysis.

Directed content analysis of the news releases was conducted by the first author. Content analysis is a method of analyzing written, verbal, or visual communication and involves characterizing these communications into manageable codes for further analysis [9]. A directed content analysis uses an existing framework to deductively apply a theory to new data [9]. Since we wanted to determine whether elements of the Strategic Risk Communication Framework were present in the news releases, a directed content analysis was the most appropriate method to achieve this goal [9]. The guiding principles of the Strategic Risk Communication Framework were used as five top-level codes: (1) decisions are evidence-based, tapping both social and natural sciences; (2) risk management and risk communication processes are transparent; (3) stakeholders are the focal point; (4) strategic risk communication is integral to integrated risk management; and (5) the strategic risk communication process requires continuous improvement through evaluation (Table 1). The first author (MF) initially read earlier news releases to become familiar with the data, then read the Strategic Risk Communication Framework to gain familiarity and understanding of the key elements. Once MF had read through the entire dataset and felt comfortable with the content, they followed the content analysis process as described by Elo and Kyngas [9]. MF completed line-by-line deductive coding, where codes were assigned to key sentences and phrases in the news releases that represented elements of the Strategic Risk Communication Framework. MF used the guiding principles of the Strategic Risk Communication Framework and created descriptions of these elements. They then read each news release, and if the text aligned with the description, codes were assigned (see Appendix A). While only one author (MF) coded the data, consistent with [9], the research team (MF and KS) continuously discussed the codes and data interpretation to increase reliability of the results. Within some of the top-level codes, the relevant content from the news releases was further characterized through inductive coding of sub-categories that better explained the data, as some top-level codes were broad and did not accurately represent the diversity of the content presented in the news releases. Sub-categories were not mutually exclusive, and quotes could be coded to multiple top-level codes and sub-categories. Sub-categories were formed inductively through reading the data, assigning key words and phrases based on the data.

## 3. Results

A total of 322 news releases (176 by the Office of the Premier and 174 by the Ministry of Health) were included in this study (Figure 1). In 2020, there was a high frequency of news releases each week, beginning in week 4 of the year. In 2021, the frequency of releases each week remained high, but was lower than in 2020. The spike in releases observed from weeks 10–17 corresponded to increased communications regarding vaccine availability for the public. Finally, 2022 showed a substantial decrease in news releases, with a maximum of 1–2 news releases per week, with many weeks containing no releases.

All elements of the Strategic Risk Communication Framework were identified across the news releases (Table 2). Overall, transparency, risk management, and stakeholders as the focal point were the most frequently identified elements across the news releases. Transparency in communications often consisted of content relating to action plans throughout the pandemic. When discussing stakeholders as the focal point, this often referred to public-centered content. Finally, many news releases discussed risk management by including statements regarding the decisions made by the government, as well as the acts, laws, and groups that were involved in those decisions. Although an important aspect of the Strategic Risk Communications Framework, the news releases did not often mention continuous improvements or corrections to earlier statements as the situation evolved.

### 3.1. Element 1: Decisions Are Evidence-Based, Tapping Both Social and Natural Sciences

Many news releases at the beginning of the pandemic discussed evidence or data supporting the decisions made by the Ontario government; however, as the pandemic progressed, this occurred less frequently. Much of the language used when describing the data that formed the basis of these decisions was vague. For example, when the need for lockdowns (i.e., closing of businesses and increasing physical distancing measures) was communicated, the rationale was “based on trends of key public health indicators” (13 July 2020); however, these indicators were not further described. Modelling of COVID-19 trends was often used as a rationale to support both why lockdowns were necessary and why lockdowns should continue to occur. Additionally, when data were provided to rationalize decisions, they often focused on data within natural sciences, with social sciences rarely being mentioned.

### 3.2. Element 2: Risk Management and Risk Communication Processes Are Transparent

Most news releases with content relating to this element focused on describing an action plan or description of the ongoing situation. Prior to the initial lockdown in March 2020, the news releases consisted of case descriptions that described the first few cases of COVID-19 in Ontario. Here, the public was provided general information on the patient, including their age, gender, location, and the treatment they were given. This content was discussed again briefly following identification of the alpha COVID-19 variant. There was also an increase in content describing vaccines beginning in November 2020, which coincided with the timing of vaccine distribution to eligible populations.

Early in the pandemic (January 2020 onwards), there was diversity in messaging in the releases, which was evident through the differences in sub-category use (Figure 2). Over time, there was a shift in the content discussed in the news releases, including a lack of diversity in messaging. Furthermore, in several instances, news releases contained quotes from speakers (e.g., Premier Doug Ford) that were used to convince the public about decisions behind their risk management actions. Occasionally when quotes were used, the language was distinct from the language used in the rest of the news release, as it involved a shift in urgency. An example of this can be seen in the news release from 17 March 2020: “This is a decision that was not made lightly. COVID-19 constitutes a danger of major proportions. We are taking this extraordinary measure because we must offer our full support and every power possible to help our health care sector fight the spread of COVID-19.”

### 3.3. Element 3: Stakeholders Are the Focal Point

Content in the news releases was frequently public-centered, with a large focus on healthcare and frontline workers at the beginning of the pandemic (Figure 3). However, as the pandemic progressed, content discussing stakeholders as the focal point diminished, with news releases from April to June 2022 not mentioning stakeholders at all. When stakeholders were mentioned, they were often accompanied by praising language, with phrases like “frontline heroes” and “Ontario spirit” being used frequently to describe healthcare workers and businesses.

There was also a strong focus on content relating to protecting the elderly and other vulnerable populations, including Indigenous people, individuals with mental health concerns, people with disabilities, and people who were homeless. Early in the pandemic, the news releases were centered on actions the government was taking to protect these vulnerable populations, for example: “We will continue to take aggressive action to support our most vulnerable residents and their caregivers” (20 April 2020). However, over time, the language shifted that responsibility to the public, as demonstrated in a news release on 19 February 2021: “As the vaccination rollout continues, it remains critically important that all Ontarians stay at home as much as possible and continue following regional public health measures, restrictions, and advice to protect our most vulnerable populations and help stop the spread of COVID-19.”

Similar to the top-level code regarding evidence-based decisions (Element 1), this code, specifically in the context of education, contained vague language that mentioned the decision but did not provide further information on how that decision was made. For example: “We are taking decisive and preventative action today to ensure students can safely return to learning in our schools” (12 April 2021).

### 3.4. Element 4: Strategic Risk Communication Is Integral to Integrated Risk Management

Content included within this code focused on higher-level descriptions of risk management processes, for example: “With a majority of Ontario adults having received a first dose of the COVID-19 vaccine and over three million doses of the Moderna vaccine arriving in June, the province is continuing to accelerate its vaccine rollout by expanding eligibility for second doses ahead of schedule.” (17 June 2021). Earlier releases more often mentioned specific laws, acts, and groups, while later releases contained more statements of decisions made by the government (Figure 4).

### 3.5. Element 5: The Strategic Risk Communication Process Requires Continuous Improvement through Evaluation

The presence of this element was the least evident across the news releases. Although it was identified in some news releases at the beginning of the pandemic, it was only used once after May 2021. An example of its use is seen in the news release on 30 May 2020: “Today, the Ontario government made amendments to the Retirement Homes Act, 2010 regulation, enabling the Retirement Homes Regulatory Authority (RHRA) to better support seniors living in retirement homes during the COVID-19 outbreak.”

There were several instances where this element could have been used in the news releases, but was not, such as in instances where something was changed but was not acknowledged. For example, in a news release from 11 December 2020, the speaker says that “…while vaccines will not be mandated during phase three, people will be strongly encouraged to get vaccinated”. However, on 17 August 2021, it is then said that “…in response to evolving data around the transmissibility of the Delta variant and based on the recent experiences of other jurisdictions, the government in consultation with the Chief Medical Officer of Health, is taking action […] this includes making COVID-19 vaccination policies mandatory in high risk settings”.

## 4. Discussion

This study aimed to determine whether communications by the Ontario government contained key elements of effective risk communication as defined by the Health Canada and Public Health Agency of Canada Strategic Risk Communication Framework. We found that, in general, news releases at the beginning of the pandemic were more comprehensive in terms of risk communication and contained most elements of the Strategic Risk Communication Framework. Additionally, earlier releases were more diverse in their content, which provided richer descriptions on each risk communication element. However, over time, the news releases became less detailed and included fewer risk communication elements. It is important to note that we cannot say whether these news releases were created with the federal government’s risk communication framework in mind. However, we can comment on whether the releases contained the key elements of the framework, and subsequently, whether any improvements in public health communication can be made.

### 4.1. Content and Language Use over Time

As the pandemic progressed, there were two notable changes in how content was portrayed across news releases. First, the news releases were most detailed at the beginning of the pandemic and contained several rich details relevant to COVID-19 (e.g., case descriptions). Second, the news releases near the beginning of the pandemic had language that displayed a sense of urgency for individuals to partake in public health measures, which was infrequently used as the pandemic progressed. Both findings may coincide with differences in how information was expressed to the public across various pandemic phases. For instance, a study evaluating global COVID-19 narratives found a quick shift in how information was communicated during “peak pandemic” compared to the “recovery phase” following the lessening of public health recommendations [10]. This communication strategy is also consistent with the need for different types of information during different pandemic phases. At the beginning of the pandemic, information most relevant to the public included timely and accurate scientific information on case descriptions and transmission mechanisms [11]. Information shared during later phases of the pandemic most often focused on vaccinations, treatments, and other actions the public could take to protect themselves and others. It should be noted that detailed pandemic information and positive risk communication has been associated with the uptake of protective behaviors by the public [12], so even as pandemic phases shift over time, detailed content should still be provided to encourage use of public health measures.

### 4.2. Language and Transparency

Vague language was frequently used across news releases when referring to the data that supported public health decisions. Vague language may be viewed by the public as a lack of transparency, which can decrease trust between the public and the government and lead to a decline in adherence to public health measures. A study of public perceptions towards COVID-19 communications by health authorities in Quebec found that the lack of transparency regarding uncertainties and evolving scientific knowledge was the most frequently identified criticism, especially surrounding the rationale to justify the implementation of public health measures [13]. In our study, a lack of transparency was further identified in instances where risk communication processes should have been evaluated and communicated (Element 5). For example, when discussing the need for mandatory vaccination policies, changes to the requirements were described but were not acknowledged. Perceived inconsistencies in messaging have been previously linked to distrust of messages and can undermine public trust in the government. For example, a study on early pandemic news coverage across Canadian media found that reporters often framed changing guidelines and lack of transparency as public health incompetence by authorities, which can damage how these changes are reported by the media [14]. Since a lack of transparency in public health messaging has also been previously reported in other studies [7,15], changes to sharing information by health authorities, including increased transparency regarding data being used to make public health decisions, are recommended.

### 4.3. Responsibility, Officials, and the Public

Our results indicate that, over time, the content described in the news releases shifted responsibilities for public health protections from the government (e.g., lockdowns) to the public (e.g., physical distancing). A similar shift from government restrictions to personal responsibility occurred in the United Kingdom in February 2022, which some researchers argued was unsustainable without the government sharing clear information about risks and providing safe environments for the public to engage in individual risk management [16]. This shift in responsibility may be viewed negatively by the public. One qualitative study found that those interviewed believed it was the government’s responsibility to create mandates and enforce the health orders that are determined by public health officials [17]. Further, these individuals believed there was a shared responsibility between the government and the public to enforce and follow public health guidelines [17]. Therefore, this shift in responsibility for public health measures from the government to the public may create a social barrier between the two parties, and by extension, cause distrust of information presented by the government [6]. While communications that invoke personal responsibility are often intended to encourage community engagement, communications that deliver imperative messages (e.g., “you should stay home”) are more effective at increasing adherence to public health measures [18]. As such, a balance between communicating imperative messages and those that invoke personal responsibilities might be needed to ensure desired adherence to public health measures.

### 4.4. Limitations

We approached our methodology (content analysis) from an interpretivist perspective, which acknowledges that each researcher inherently influences data interpretation and analysis, and as such, there may be differences between coders [19]. Since only one person coded the data, differences may be expected if another researcher was to conduct the analysis. However, this choice was made to allow the single coder to become immersed in the data rather than verify accuracy across coders [9]. Since our analysis relates to only one framework that is not heavily present in the previously published literature, this may limit the generalizability of our findings. However, the elements of the framework are not unique, and are present in many other risk communication frameworks, such as the CERC framework. Our study focused on only one source of public health communications (the news releases), so further research is recommended to investigate content of other sources and how they may influence perceptions of public health messaging.

## 5. Conclusions

This study evaluated whether the COVID-19 communications by the Ontario government contained key elements of effective risk communication, as defined by the Strategic Risk Communication Framework, and whether improvements in risk communication could be made. Our findings identified several areas where risk communication could be improved, such as increasing transparency regarding evidence-based decisions and explaining rationales behind changes in decisions. Based on our findings, there are several practical recommendations for the Ontario government. We recommend increasing transparency regarding the evidence and rationale behind public health decisions to improve public trust, and, therefore, compliance in health measures. We further recommend that communications are detailed whenever possible and are frequently improved upon to effectively convey the level of risk to the public and promote protective measures throughout all pandemic phases. Adhering to best practices for risk communication will allow the public to make informed decisions about their health and take the necessary measures to stay safe.

## Figures and Tables

**Figure 1 ijerph-21-00351-f001:**
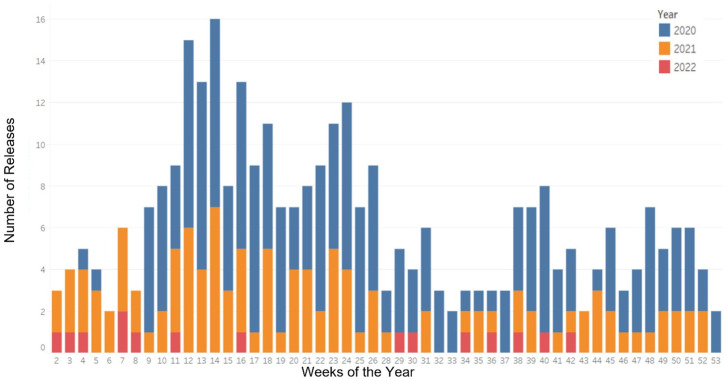
Frequency of COVID-19 news releases by the Ontario government, 2020–2022, by week.

**Figure 2 ijerph-21-00351-f002:**
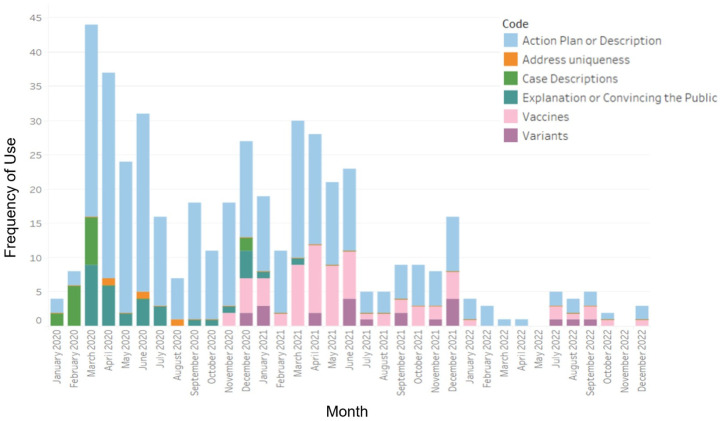
Frequency of coding for sub-categories of the element “Risk management and risk communication processes are transparent”, by month, from January 2020 to December 2022.

**Figure 3 ijerph-21-00351-f003:**
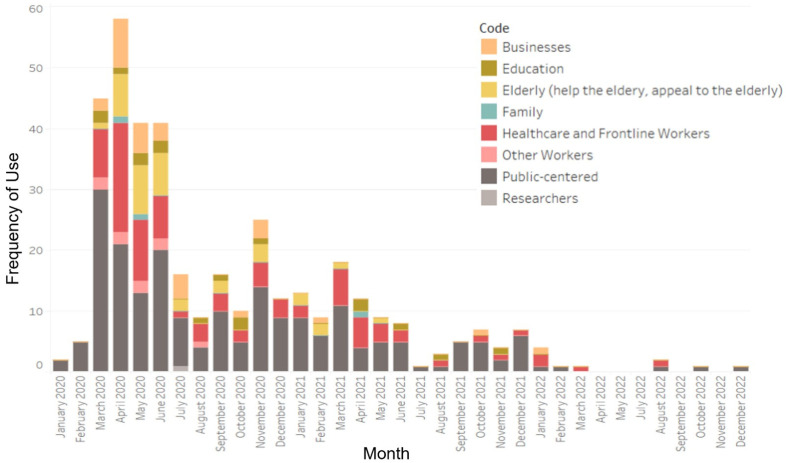
Frequency of coding for sub-categories of the element “Stakeholders are the focal point”, by month, from January 2020 to December 2022.

**Figure 4 ijerph-21-00351-f004:**
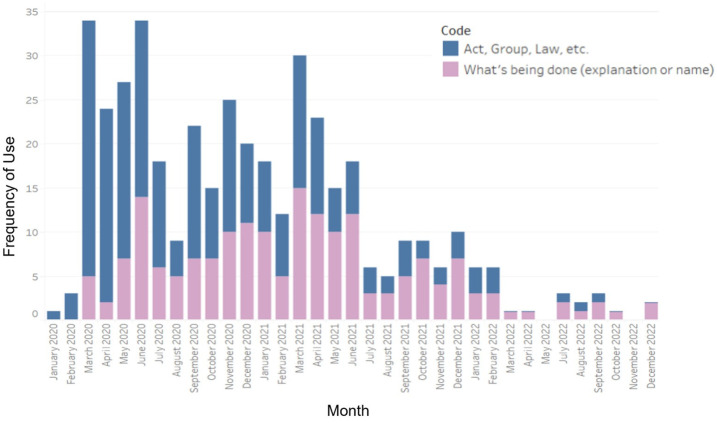
Frequency of coding for sub-categories of the element “Strategic risk communication is integral to integrated risk management”, by month, from January 2020 to December 2022.

**Table 1 ijerph-21-00351-t001:** Description of the elements of the Strategic Risk Communications Framework, which was used in the directed content analysis of Ontario government news releases on COVID-19. Descriptions adapted from those provided in the Strategic Risk Communications Framework [3].

Element	Description	Sub-Category Classification
Decisions are evidence-based, tapping both social and natural sciences	Used to describe instances where decisions that were made were based on, and supported by, existing data.	Not applicable
Risk management and risk communication processes are transparent	Used to describe transparency regarding the action being taken by the government. When facts are unknown, the communicator must be clear about the gaps remaining and what efforts are being taken to fill them.	Action plan or description, address uniqueness, case descriptions, explanation or convincing the public, vaccines, variants
Stakeholders are the focal point	Used when addressing or discussing protection of a specific group of people, the public, or certain language is used to appeal to a specific group of people.	Businesses, education, elderly (helping and appealing to), family, healthcare and frontline workers, other workers, public-centered, researchers
Strategic risk communication is integral to integrated risk management	Used when mentioning specific groups or laws at an organizational level to manage risk, as well as a brief statements regarding how risk is being managed.	Act, group, law, etc., what is being executed (explanation or name)
The strategic risk communication process requires continuous improvement through evaluation	Used when a previously created plan was revised, and this was acknowledged.	Not applicable

**Table 2 ijerph-21-00351-t002:** Exemplar quotes illustrating occurrences of each Strategic Risk Communication Framework element.

Element	Quote ^1^	Frequency of Code Use ^2^
Decisions are evidence-based, tapping both social and natural sciences	“As Ontario continues down the path to economic recovery, decisions on which regions will enter Stage 3 and when will be made in consultation with the Chief Medical Officer of Health and other health experts and **based on trends of key public health indicators**.” (13 July 2020)	102
Risk management and risk communication processes are transparent	“**Due to the continuing success of Ontario’s vaccine rollout** and the collective efforts of Ontarians in following public health and workplace safety measures to date, effective 22 May 2021 at 12:01 a.m. the province will reopen outdoor recreational amenities with restrictions in place, such as the need to maintain physical distancing.” (20 May 2021)	914
Stakeholders are the focal point	“As the COVID-19 outbreak continues to evolve globally, Ontario is taking further action to ensure the province’s health care system is positioned to **continue to safeguard the health and wellbeing of Ontarians**.” (12 March 2020)	722
Strategic risk communication is integral to integrated risk management	“To support Ontarians as they begin to safely plan for the season, the Ontario government, **based on the advice of the Chief Medical Officer of Health and input from the Public Health Measures Table**, is providing preliminary guidance on how to safely celebrate this year and protect your loved ones.” (25 November 2020)	836
The strategic risk communication process requires continuous improvement through evaluation	“**To do so, the province is amending O. Reg. 272/21 under the Emergency Management and Civil Protection Act (EMCPA)** to ensure patients receive quality care in the most appropriate setting during the third wave of the pandemic, driven by variants of concern.” (28 April 2021)	36

^1^ Emphasis (bold) added to demonstrate which part of the quote was coded to the element. ^2^ A total of 322 news releases were coded. Multiple codes can be included in a single news release, and sentences can be coded under multiple elements.

## Data Availability

All data supporting the results of this article are publicly available from https://news.ontario.ca/en (accessed on 14 March 2024).

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
