# Peer review of "Content Analysis of Official Public Health Communications in Ontario, Canada during the COVID-19 Pandemic"

_ijerph, 2024, doi:10.3390/ijerph21030351_

Round 1
Reviewer 1 Report
Comments and Suggestions for Authors
Dear authors,
Firstly, I would like to express my appreciation for your work. I recognize that the context within which you have conducted your research has not been extensively explored, and undertaking such a study demands considerable effort and precision. I particularly enjoyed the manuscript's language, finding it easy to follow, read, and comprehensive. While I prefer succinct paragraphs, I do not consider this a critical issue.
Upon reviewing your work, I have identified several concerns that I believe could further enhance the quality of the manuscript:
Introduction
In lines 68-69, you mention the use of a handbook. It is unclear whether all aspects of the handbook were followed or if certain criteria or codes were specifically investigated. Clarifying this could help narrow the research framework and better convey the hypothesis to readers. For example, in my examination of the "Strategic Risk Communications Framework for Health Canada and the Public Health Agency of Canada," a 26-page document, I noted that the guideline provides specific points for each category. It would be beneficial to specify which points and aspects of this guideline were investigated in your study. Although this is mentioned in the methods section, including a brief description in the hypothesis section would enhance clarity.
Materials and Methods
- Line 74: Is the web address mentioned as the sole public engagement tool used by the government, or are there others? If it is the only one, this comment can be disregarded. Otherwise, it would be useful to know why only this platform was analyzed and whether other public media utilized by the government were considered.
- Line 78: A brief explanation of why this particular time window was chosen would aid readers in understanding the purpose and scope of the study.
- Line 79: When mentioning the use of "NVivo v1.7.1" for analysis, it would be preferable to include a citation for the tool, if possible.
- Line 92: The process of coding and the confidence in the assigned codes could be elaborated upon. How do you ensure the accuracy and appropriateness of the coding? It would be helpful if any supporting materials or frameworks, such as those by Elo & Kyngäs (2008), were provided for reference. Additionally, more detail on the initial reading and analysis process by the first author could improve understanding and replicability.
- Table 1 presents a challenge in readability due to the close spacing of descriptions in the PDF version I reviewed. Adding borders or adjusting the formatting to enhance readability would be beneficial.
Results
I found the presentation of results satisfactory. Given the high number of records, it is understandable that detailing every aspect might not be feasible. However, if you are willing to share more detailed results, including them as an appendix could be invaluable. This suggestion stems from the observation that work in this area is not widely represented in both experiments and literature. Sharing detailed results could significantly aid future research in this field.
Author Response
Dear Reviewer 1,
Thank you very much for your comments and the time you have taken to review this manuscript. Please see our point-by-point response to your comments below. We have also uploaded a revised manuscript with tracked changes to see where we have made our revisions.
Response to Reviewer:
In lines 68-69, you mention the use of a handbook. It is unclear whether all aspects of the handbook were followed or if certain criteria or codes were specifically investigated. Clarifying this could help narrow the research framework and better convey the hypothesis to readers. For example, in my examination of the "Strategic Risk Communications Framework for Health Canada and the Public Health Agency of Canada," a 26-page document, I noted that the guideline provides specific points for each category. It would be beneficial to specify which points and aspects of this guideline were investigated in your study. Although this is mentioned in the methods section, including a brief description in the hypothesis section would enhance clarity.
RESPONSE: We used the Guiding Principles of the Health Canada and Public Health Agency of Canada. These guiding principles outlined 5 key elements that were essential for risk communication between the government and the public. We have added these details to the section to more clearly demonstrate which aspects of the handbook were followed.
Materials and Methods
- Line 74: Is the web address mentioned as the sole public engagement tool used by the government, or are there others? If it is the only one, this comment can be disregarded. Otherwise, it would be useful to know why only this platform was analyzed and whether other public media utilized by the government were considered.
RESPONSE: The news releases used in this study were not the sole public engagement tool used by the government, since they used other media sources (e.g., YouTube and social media). We decided to use the news releases as they were the official communications from the government and were regularly updated prior to messages being shared using other media sources. We have added further details as to why this website was chosen in our methods section.
- Line 78: A brief explanation of why this particular time window was chosen would aid readers in understanding the purpose and scope of the study.
RESPONSE: The rationale for the particular time window follows that January 25, 2020 was the first news release regarding COVID-19, and December 15, 2022 was the final news release regarding COVID-19 that took place in 2022. After 2022, there were so few news releases that we felt comfortable ending our analysis here. The methods have been updated to reflect this.
- Line 79: When mentioning the use of "NVivo v1.7.1" for analysis, it would be preferable to include a citation for the tool, if possible.
RESPONSE: While a standard citation format for NVivo software does not seem to be available, we added further information about the software and publisher in the methods section.
- Line 92: The process of coding and the confidence in the assigned codes could be elaborated upon. How do you ensure the accuracy and appropriateness of the coding? It would be helpful if any supporting materials or frameworks, such as those by Elo & Kyngäs (2008), were provided for reference. Additionally, more detail on the initial reading and analysis process by the first author could improve understanding and replicability.
RESPONSE: Thank you for this comment. The first author (MF) initially read earlier news releases to become familiar with the data, then read the Strategic Risk Communication framework to gain familiarity and understanding of the key elements. Once MF had read through the entire dataset and felt comfortable with the content, they followed the content analysis process as described by Elo & Kyngas [9]. MF completed line-by-line deductive coding, where codes were assigned to key sentences and phrases in the news releases that represented elements of the Strategic Risk Communication Framework. MF used the Guiding Principles of the Strategic Risk Communication Framework and created descriptions of these elements. They then read each news release, and if the text aligned with the description, codes were assigned. While only one author (MF) coded the data, consistent with [19], the research team (MF and KS) continuously discussed the codes and data interpretation to increase reliability of the results. Within some of the top-level codes, the relevant content from the news releases was further characterized through inductive coding of sub-categories that better explained the data, as some top-level codes were broad and did not accurately represent the diversity of the content presented in the news releases. Sub-categories were not mutually exclusive, and quotes could be coded to multiple top-level codes and sub-categories. Sub-categories were formed inductively through reading the data assigning key words and phrases based on the data. We have added this rationale into the manuscript as well. Additionally, we discuss further the potential limitations of our approach in the discussion section.
- Table 1 presents a challenge in readability due to the close spacing of descriptions in the PDF version I reviewed. Adding borders or adjusting the formatting to enhance readability would be beneficial.
RESPONSE: We have expanded the table boundaries and revised the formatting to improve readability of both tables.
Results
I found the presentation of results satisfactory. Given the high number of records, it is understandable that detailing every aspect might not be feasible. However, if you are willing to share more detailed results, including them as an appendix could be invaluable. This suggestion stems from the observation that work in this area is not widely represented in both experiments and literature. Sharing detailed results could significantly aid future research in this field.
RESPONSE: Thank you for this comment. All data supporting our analyses are publicly available, but to increase visibility and ease of continuing work in this area, we will upload the direct links to the news releases as supplementary materials. Additionally, we will upload a copy of the codebook that supports our results.
Reviewer 2 Report
Comments and Suggestions for Authors
Dear Respected Authors
Thank you for considering a significant area of research related to the COVID-19 pandemic. The authors conducted a content analysis to determine whether news releases by the Ontario government contained key elements of effective risk communication, as outlined by the Health Canada and Public Health Agency of Canada Strategic Risk Communication Framework. The results are interesting but some comments need more clarification.
- Lines 77-78, why did you select this period? Is there a rationale behind this or not?
- What other things have you done to increase your study rigor/consistency? for example memos etc.
- Line 213, this statement does not need a reference.
- Based on these results, what is your practical recommendation for the Ontario government for future pandemics like COVID-19? or what are the lessons learned from this study?
- What is your rationale for using Direct CA? Please list some reasons that show this approach fits your synthesis.
Cheers
Author Response
Dear Reviewer 2,
Thank you very much for your comments and the time you have taken to review this manuscript. Please see our point-by-point response to your comments below. We have also uploaded a revised manuscript with tracked changes to see where we have made our revisions.
Response to Reviewer:
Lines 77-78, why did you select this period? Is there a rationale behind this or not?
RESPONSE: The rationale for the particular time window follows that January 25, 2020 was the first news release regarding COVID-19, and December 15, 2022 was the final news release regarding COVID-19 that took place in 2022. After 2022, there were so few news releases that we felt comfortable ending our analysis here. The methods have been updated to reflect this.
What other things have you done to increase your study rigor/consistency? for example memos etc.
RESPONSE: Thank you for this comment. The first author (MF) initially read earlier news releases to become familiar with the data, then read the Strategic Risk Communication framework to gain familiarity and understanding of the key elements. Once MF had read through the entire dataset and felt comfortable with the content, they followed the content analysis process as described by Elo & Kyngas [9]. MF completed line-by-line deductive coding, where codes were assigned to key sentences and phrases in the news releases that represented elements of the Strategic Risk Communication Framework. MF used the Guiding Principles of the Strategic Risk Communication Framework and created descriptions of these elements. They then read each news release, and if the text aligned with the description, codes were assigned. While only one author (MF) coded the data, consistent with [19], the research team (MF and KS) continuously discussed the codes and data interpretation to increase reliability of the results. Within some of the top-level codes, the relevant content from the news releases was further characterized through inductive coding of sub-categories that better explained the data, as some top-level codes were broad and did not accurately represent the diversity of the content presented in the news releases. Sub-categories were not mutually exclusive, and quotes could be coded to multiple top-level codes and sub-categories. Sub-categories were formed inductively through reading the data assigning key words and phrases based on the data. We have added this rationale into the manuscript as well. Additionally, we discuss further the potential limitations of our approach in the discussion section.
Line 213, this statement does not need a reference.
RESPONSE: We have removed the reference.
Based on these results, what is your practical recommendation for the Ontario government for future pandemics like COVID-19? or what are the lessons learned from this study?
RESPONSE: Thank you for this comment. Based on our findings, there are several practical recommendations for the Ontario government. We recommend increasing transparency regarding the evidence and rationale behind public health decisions to improve public trust, and therefore, compliance in health measures. We further recommend that communications are detailed whenever possible and are frequently improved on to effectively convey the level of risk to the public and promote protective measures throughout all pandemic phases. We have expanded on this within the conclusions.
What is your rationale for using Direct CA? Please list some reasons that show this approach fits your synthesis.
RESPONSE: Since we wanted to determine whether elements of the Strategic Risk Communication Framework were present in the news releases, a directed content analysis was the most appropriate method to achieve this goal [9]. The guiding principles of the Strategic Risk Communication Framework were used as five top-level codes. We have added this rationale within the methods section.
Reviewer 3 Report
Comments and Suggestions for Authors
The paper conducts a content analysis of public health communications in Ontario, Canada, during the COVID-19 pandemic. It assesses if these communications contain key elements of effective risk communication outlined by the Health Canada and Public Health Agency of Canada Strategic Risk Communication Framework. The study finds that while all communications contained at least one framework element, the presence of these elements varied over time. Early in the pandemic, communications were more comprehensive, but over time, they became less detailed and utilized more vague language. The paper recommends increasing transparency regarding evidence-based decisions and explaining decision changes to improve risk communication.
Drawbacks:
1. The use of vague language and a lack of transparency in explaining public health decisions could undermine public trust and adherence to health measures.
2. Communications became less detailed over time, potentially reducing their effectiveness in conveying necessary public health information.
3. The analysis noted a lack of continuous improvements or corrections to earlier statements, which is crucial for adapting to evolving public health scenarios.
Recommendations:
1. Clearly articulate the evidence and rationale behind public health decisions to improve public trust and compliance.
2. Ensure communications remain detailed and diverse to effectively convey risk and promote protective behaviors throughout all pandemic phases.
3. Incorporate and communicate continuous improvements in public health strategies and messaging to adapt to the changing pandemic landscape and enhance public engagement.
Author Response
Dear Reviewer 3,
Thank you very much for your comments and the time you have taken to review this manuscript. We have uploaded a revised manuscript with tracked changes to see where we have made our revisions. We have also provided a point-by-point response to your comments below.
In addition to the below improvements, we have also addressed the comments in your overall review report, including (1) adding clarifications to improve the reporting of our methods, (2) providing additional details on our results in supplementary materials, and (3) improving the reporting of our conclusions.
Response to Reviewer:
The paper conducts a content analysis of public health communications in Ontario, Canada, during the COVID-19 pandemic. It assesses if these communications contain key elements of effective risk communication outlined by the Health Canada and Public Health Agency of Canada Strategic Risk Communication Framework. The study finds that while all communications contained at least one framework element, the presence of these elements varied over time. Early in the pandemic, communications were more comprehensive, but over time, they became less detailed and utilized more vague language. The paper recommends increasing transparency regarding evidence-based decisions and explaining decision changes to improve risk communication.
Drawbacks:
1. The use of vague language and a lack of transparency in explaining public health decisions could undermine public trust and adherence to health measures.
2. Communications became less detailed over time, potentially reducing their effectiveness in conveying necessary public health information.
3. The analysis noted a lack of continuous improvements or corrections to earlier statements, which is crucial for adapting to evolving public health scenarios.
Recommendations:
1. Clearly articulate the evidence and rationale behind public health decisions to improve public trust and compliance.
2. Ensure communications remain detailed and diverse to effectively convey risk and promote protective behaviors throughout all pandemic phases.
3. Incorporate and communicate continuous improvements in public health strategies and messaging to adapt to the changing pandemic landscape and enhance public engagement.
RESPONSE: Thank you for your comments. We are happy to see that your interpretation of the main conclusions was consistent with our reporting. We have improved our presentation of the main findings and the recommendations in the Discussion section.